# LEARNING AN OBJECT BASED MEMORY SYSTEM

## ABSTRACT

A robot operating in a household makes observations of multiple objects as it moves around over the course of days or weeks. The objects may be moved by inhabitants, but not completely at random. The robot may be called upon later to retrieve objects and will need a long-term object-based memory in order to know how to find them. In this paper, we combine some aspects of classic techniques for data-association filtering with modern attention-based neural networks to construct object-based memory systems that consume and produce high-dimensional observations and hypotheses. We perform end-to-end learning on labeled observation trajectories to learn both the internal transition and observation models. We demonstrate the system's effectiveness on a sequence of problem classes of increasing difficulty and show that it outperforms clustering-based methods, classic filters, and unstructured neural approaches.

## 1 INTRODUCTION

Consider a robot operating in a household, making observations of multiple objects as it moves around over the course of days or weeks. The objects may be moved by the inhabitants, even when the robot is not observing them, and we expect the robot to be able to find any of the objects when requested. We will call this type of problem *entity monitoring*. It occurs in many applications, but we are particularly motivated by the robotics applications where the observations are very high dimensional, such as images or point clouds. Such systems need to perform online *data association*, determining which individual objects generated each observation, and *state estimation*, aggregating the observations of each individual object to obtain a representation that is lower variance and more complete than any individual observation. This problem can be addressed by an online recursive *filtering* algorithm that receives a stream of object detections as input and generates, after each input observation, a set of hypotheses corresponding to the actual objects observed by the agent.

When observations are closely spaced in time and objects only briefly go out of view, the entity monitoring problem becomes the well studied problem of *object tracking*. In contrast, in this paper, we are interested in studying the more generalized entity monitoring problem, where a robot must associate a set of sparse and temporally separated observations of objects over the course of days or weeks into a coherent estimate of the underlying objects and associated properties in a scene (Figure 1). In such a setting, it is important that the system does not depend on continuous visual tracking, as any individual object may be seen at one time and then again significantly later. A sub-problem of generalized entity monitoring corresponds to *object identification*, in which we seek to consistently re-identify objects across time. However, to solve the generalized entity monitoring problem, a system must not only identify similar objects across time, but integrate the observations into an estimate of their *properties* that may not be directly inferrable from any single observation.

A classical solution to the entity monitoring problem, developed for the tracking case but extensible to other dynamic settings, is a *data association filter* (DAF) (the tutorial of Bar-Shalom et al. (2009) provides a good introduction). A Bayes-optimal solution to this problem can be formulated, but it requires representing a number of possible hypotheses that grows exponentially with the number of observations. A much more practical, though less robust, approach is a maximum likelihood DAF (ML-DAF), which commits, on each step, to a maximum likelihood data association: the algorithm maintains a set of object hypotheses, one for each object (generally starting with the empty set) and for each observation it decides to either: (a) associate the observation with an existing object hypothesis and perform a Bayesian update on that hypothesis with the new data, (b) start a new object hypothesis based on this observation, or (c) discard the observation as noise. As the number of entities

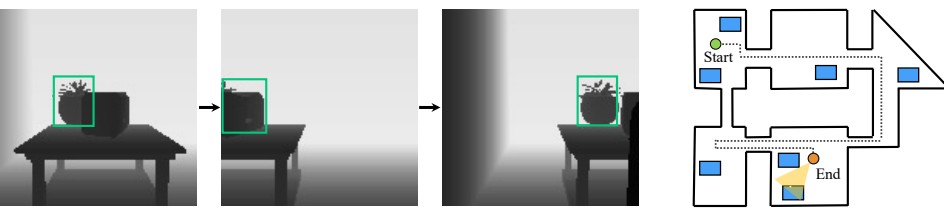

Observations (One Segmented Depth Map per Timestep)      House Plan

Figure 1: **(left) Example observations to the** OBM **system.** At each time-step, the OBM obtains a segmented depth map of a single object. **(right) Example domain layout.** Sample layout with a robot trajectory, field of view (in yellow) and tables that can contain objects. Objects in the domain can move both locally on the table they are on as well as to different tables (simulating perturbations induced by the inhabitants). The robot moves through the environment acquiring local, partial observations of objects and must predict the number, location, and shape of objects it has seen.

in the domain and the time between observations of the same entity increase, the problem becomes more difficult and the system can begin to play the role of the long-term object-based memory (OBM) for an autonomous agent.

The engineering approach to constructing such an OBM requires many design choices, including the specification of a latent state-space for object hypotheses, a model relating observations to object states, another model specifying the evolution of object states over time, and thresholds or other decision rules for choosing, for a new observation, whether to associate it with an existing hypothesis, use it to start a new hypothesis, or discard it. In any particular application, the engineer must tune all of these models and parameters to build an OBM that performs well. This is a time-consuming process that must be repeated for each new application.

In this paper, we develop a method for training neural networks to perform as OBMs for dynamic entity monitoring. *In particular, we train a system to construct a memory of the objects in the environment, without explicit models of the robot's sensors, the types of objects to be encountered, or the patterns in which they might move in the environment.* Although it is possible to train an unstructured recurrent neural network (RNN) to solve this problem, we find that building in some aspects of the structure of the OBM allows faster learning with less data and enables the system to address problems with a longer horizon. We describe a neural-network architecture that uses self-attention as a mechanism for data association, and demonstrate its effectiveness in several illustrative problems. We first validate that OBM can estimate object states when observations are drawn online from a set of cluster centers. Next, we validate that OBM can estimate object states when observations correspond to high-dimensional images. Finally, we illustrate its application on a realistic simulated robotic domain.

## 2 RELATED WORK

**Online clustering methods** In the simple setting, where object state does not change over time, the entity monitoring problem can be seen as a form of online clustering, where the assignment of data points to clusters is done *online*, with observations arriving sequentially and a cumulative set of hypotheses output after each observation. One of the most fundamental online clustering methods is *vector quantization*, articulated originally by Gray (1984) and understood as a stochastic gradient method by Kohonen (1995). It initializes cluster centers at random and assigns each new observation to the closest cluster center, and updates that center to be closer to the observation. We show that our approach can learn to outperform this online clustering method. More recent work has explored theoretical aspects of online clustering with guarantees (Liberty et al., 2016; Bhaskara and Rwanpathirana, 2020; Cohen-Addad et al., 2021).

**Data-Association Filters** The most classic filter, for the case of a single entity, is the Kalman filter (Welch and Bishop, 2006). In the presence of data-association uncertainty the Kalman filter can be extended by considering assignments of observations to multiple existing hypotheses in a DAF or ML-DAF. These approaches, all of which require hand-tuned transition and observation models, are described by (Bar-Shalom et al., 2009). We show that our approach can learn the underlying transition and observation models and performs comparably to ML-DAF with ground truth system dynamic and observation models.

**Visual data-association methods** A special case of the entity monitoring problem where observations are closely spaced in time has been extensively explored in the visual object tracking setting (Luo et al., 2014; Xiang et al., 2015; Bewley et al., 2016; Frossard and Urtasun, 2018; Brasó and

Leal-Taixé, 2020). In these problems, there is typically a fixed visual field populated with many smoothly moving objects. This enables some specialized techniques that take advantage of the fact that the observations of each object are typically smoothly varying in space-time, and incorporate additional visual appearance cues. In contrast, in our setting, there is no fixed spatial field for observations and they may be temporally widely spaced, as would be the case when a robot moves through the rooms of a house, encountering and re-encountering different objects as it does so. While work has studied the detection of repeated objects with similar appearance (Girdhar and Ramanan, 2019; Bai et al., 2019; Bansal et al., 2021; He et al., 2021; Zhang et al., 2021b;a; Huang et al., 2019), our focus is on aggregating and estimating the individual states of objects based on substantially different observations in a different space, and our methods are not competitive with specialized techniques on the much more specialized problems of fixed-visual-field tracking or object re-identification.

**Learning for data association**     There is relatively little work in the area of learning for generalized data association, but Liu et al. (2019) provide a recent application of LSTMs (Hochreiter and Schmidhuber, 1997) to a rich version of the data association problem, in which batches of observations arrive simultaneously, with a constraint that each observation can be assigned to at most one object hypothesis. The sequential structure of the LSTM is used here not for recursive filtering, but to handle the variable numbers of observations and hypotheses. It is assumed that Euclidean distance is an appropriate metric and that the observation and state spaces are the same. Milan et al. (2017) combine a similar use of LSTM for data association with a recurrent network that learns to track multiple targets. It learns a dynamics model for the targets, including birth and death processes, but operates in simple state and observation spaces.

**Slot Based and Object Centric Learning**     Our approach to the dynamic entity monitoring task relies on the use of attention over a set of object hypothesis slots. Generic architectures for processing such slots can be found in (Shi et al., 2015; Vinyals et al., 2015; Lee et al., 2018), where we use (Lee et al., 2018) as a point of comparison for OBM. We note that these architectures provide generic mechanisms to process sets of inputs, and lack the explicit structure from DAF we build into our model. Our individual hypothesis slots correspond to beliefs over object hypotheses, and thus relates to existing work in object-centric scene learning. Such work has explored the discovery of factorized objects from both static scenes (Burgess et al., 2019; Greff et al., 2019; Locatello et al., 2020). Developed concurrently and most similar to our work, (Locatello et al., 2020) also utilizes slots as a means of representing a factorized object decomposition of static images. In contrast to (Locatello et al., 2020), our work focuses on the use of a set of slots to represent the evolution of uncertain object hypotheses over time, and incorporates attention and inductive biases from DAF to selectively update beliefs across time to obtain object hypotheses as well as their associated confidences.

**Algorithmic priors for neural networks**     One final comparison is to other methods that integrate algorithmic structure with end-to-end neural network training. This approach has been applied to sequential decision making by Tamar et al. (2016), particle filters by Jonschkowski et al. (2018), and Kalman filters by Krishnan et al. (2015), as well as to a complex multi-module robot control system by Karkus et al. (2019). The results generally are much more robust than completely hand-built models and much more sample-efficient than completely unstructured deep-learning. We view our work as an instance of this general approach.

## 3   PROBLEM FORMULATION

We formalize the process of learning an object-based memory system (OBM). Formally, when the OBM is executed online, it receives a stream of input observations $z_1, \ldots z_T$ where $z_t \in \mathbb{R}^{d_z}$, and after each input $z_t$, it will output two vectors representing a set of predicted properties of hypothesized objects $y_t = [y_{tk}]_{k \in (1..K)}$ and an associated confidence score for each hypothesis, $c_t = [c_{tk}]_{k \in (1..K)}$, where $y_{tk} \in \mathbb{R}^{d_y}$, $c_{tk} \in (0, 1)$. To ensure that confidences are bounded, we constrain $\sum_k c_{tk} = 1$. We limit the maximum number of hypothesis "slots" in advance to $K$. Dependent on the application, the $z$ and $y$ values may be in the same space with the same representation, but this is not necessary.

We have training data representing $N$ different entity-monitoring problem instances, $\mathcal{D} = \{(z_t^{(i)}, m_t^{(i)})_{t \in (1..L_i)}\}_{i \in (1..N)}$, where each training example is an input/output sequence of length $L_i$, each element of which consists of a pair of input $z$ and $m = \{m_j\}_{j \in (1..J_t^{(i)})}$, which is a set of nominal object hypotheses representing the true *current state* of objects that have actually been observed so far in the sequence. It will always be true that $m_t^{(i)} \subseteq m_{t+1}^{(i)}$ and $J_t^{(i)} \leq K$ because the set of objects seen so far is cumulative.

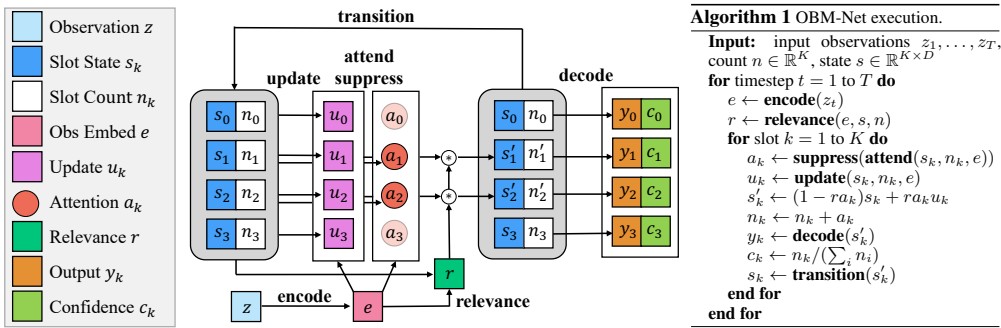

Figure 2: Architecture and pseudocode of OBM-Net. Observations are fed sequentially to OBM-Net, and encoded with respect to each hypothesis. A subset of the hypotheses are updated at each time-step, with corresponding slot counts incremented according to attention weight. Slots are then decoded, with the confidence of an output proportional to underlying slot count.

Our objective is to train a recurrent computational model to perform as an OBM effectively in problems that are drawn from the same distribution over latent domains as those in the training set. To do so, we formulate a model (described in section 4) with parameters $\theta$, which transduces the input sequence $z_1, \ldots, z_L$ into an output sequence $(y_1, c_1), \ldots, (y_L, c_L)$, and train it to minimize the following loss function:

$$\mathcal{L}(\theta; \mathcal{D}) = \sum_{i=1}^{N} \sum_{t=1}^{L_i} \mathcal{L}_{\text{obj}}(y_t^{(i)}, m_t^{(i)}) + \mathcal{L}_{\text{slot}}(y_t^{(i)}, c_t^{(i)}, m_t^{(i)}) + \mathcal{L}_{\text{sparse}}(c_t^{(i)}) \ . \tag{1}$$

The $\mathcal{L}_{\text{obj}}$ term is a *chamfer loss* (Barrow et al., 1977), which looks for the predicted $y_k$ that is closest to each actual $m_j$ and sums their distances, making sure the model has found a good, high-confidence representation for each true object, with $\epsilon \ll 1$ :

$$\mathcal{L}_{\text{obj}}(y, c, m) = \sum_{j} \min_{k} \frac{1}{c_k + \epsilon} \|y_k - m_j\| \ .$$

The $\mathcal{L}_{\text{slot}}$ term is similar, but makes sure that each object the model has found is a true object, where we multiply by $c_k$ to not penalize for predicted objects in which we have low confidence:

$$\mathcal{L}_{\text{slot}}(y, c, m) = \sum_{k} \min_{j} c_k \|y_k - m_j\| \ .$$

Finally, the sparsity loss discourages the model from using multiple outputs to represent the same true object, by encouraging sparsity in object hypothesis confidences (derivation in Section D):

$$\mathcal{L}_{\text{sparse}}(c) = -\log\|c\| \ .$$

## 4 OBM-NETS

Inspired by the the basic form of classic DAF algorithms and the ability of modern neural-network techniques to learn complex models, we have designed the OBM-Net architecture for learning OBMs and a cuwwwstomized procedure for training it from data, motivated by several design considerations. First, because object hypotheses must be available after each individual input and because observations will generally be too large and the problem too difficult to solve from scratch each time, the network will have the structure of a recursive filter, with new memory values computed on each observation and then fed back for the next. Second, because the loss function is *set based*, that is, it doesn't matter what order the object hypotheses are delivered in, our memory structure should also be permutation invariant and independent of the number of objects, and so the memory processing is in the style of an attention mechanism. Finally, in applications where the observations $z$ may be in a representation not well suited for hypothesis representation and aggregation, the memory operates on a latent representation that is related to observations and output hypotheses via encoder and decoder modules.

Figure 2 shows the architecture of the OBM-Net model. The memory of the system is stored in $s$, which consists of $K$ elements, the $K$ hypotheses in DAF, each in $\mathbb{R}^{d_s}$; the length-$K$ vector $n$ of positive values encodes how many observations have been assigned to each slot during the execution so far. New observations are combined with the memory state, and the state is updated to reflect the passage of time by a neural network constructed from seven modules with trainable weights.

When an observation $z$ arrives, it is immediately **encode**d into a vector $e$ in $\mathbb{R}^{d_s}$, which is fed into subsequent modules. First, **attention** weights $w$ are computed for each hypothesis slot, using the

encoded input and the existing content of that slot, representing the degree to which the current input "matches" the current value of each hypothesis in memory (corresponding to hypothesis matching computation in DAF algorithms). To force the network to commit to a sparse assignment of observations to object hypotheses while retaining the ability to effectively train with gradient descent, the **suppress** module sets all but the top $M$ values in $w$ to 0 and renormalizes, to obtain the vector $a$ of $M$ values that sum to 1:

$$w_k = \frac{\exp(\mathbf{attend}(s_k, n_k, e))}{\sum_{j=0}^{n} \exp(\mathbf{attend}(s_j, n_k, e))} \;\; ; \;\; a = \mathbf{suppress}(w) \;\; .$$

The $a$ vectors are integrated to obtain $n$, which is normalized to obtain the output confidence $c$.

The **update** module also operates on the encoded input and the contents of each hypothesis slot, producing a hypothetical update of the hypothesis in that slot under the assumption that the current $z$ is an observation of the object represented by that slot (corresponding to hypothesis updates in DAF algorithms); so for all slots $k$,

$$u_k = \mathbf{update}(s_k, n_k, e) \;\; .$$

Additionally, a scalar **relevance** value, $r \in (0, 1)$, is computed from $s$ and $e$; this value modulates the degree to which slot values are updated, and gives the machine the ability to ignore or downweight an input, corresponding to rejection of outlier observations in DAF algorithms. It is computed as

$$r = \mathbf{relevance}(e, s, n) = \mathrm{NN}_2(\underset{k=1}{\overset{K}{\mathrm{avg}}} \, \mathrm{NN}_1(e, s_k, n_k)) \;\; ,$$

where $\mathrm{NN}_1$ is a fully connected network with the same input and output dimensions and $\mathrm{NN}_2$ is a fully connected network with a single sigmoid output unit. The attention output $a$ and relevance $r$ are now used to decide how to combine all possible slot-updates $u$ with the old slot values $s_t$ using the following fixed formula for each slot $k$:

$$s'_{tk} = (1 - ra_k)s_{tk} + ra_k u_k \;\; .$$

Because most of the $a_k$ values have been set to 0, this results in a sparse update which will ideally concentrate on a single slot to which this observation is being "assigned", and correspond to the DAFhypothesis updates in DAF algorithms.

To obtain outputs, slot values $s'_t$ are then **decoded** into the outputs, $y$, using a fully connected network:

$$y_k = \mathbf{decode}(s'_{tk}) \;\; .$$

Finally, to handle the setting in which object state evolves over time, we add a **transition** module, which computes the state $s_{t+1}$ from the new slot values $s'_t$ using an additional neural network, corresponding to DAFtransition updates in DAF algorithms:

$$s_{t+1\,k} = \mathbf{transition}(s'_t)_k \;\; .$$

These values are then fed back, recurrently, as inputs to the overall system.

Given a data set $\mathcal{D}$, we train the OBM-Net model end-to-end to minimize loss function $\mathcal{L}$, with a slight modification. We find that including the $\mathcal{L}_{\mathrm{sparse}}$ term from the beginning of training results in poor learning, but adopting a training scheme in which the $\mathcal{L}_{\mathrm{sparse}}$ is first omitted then reintroduced over training epochs, results in reliable training that is efficient in both time and data.

## 5 EMPIRICAL RESULTS

We evaluate OBM-Net on several different *entity monitoring* tasks. First, we consider a simple online clustering task and validate the underlying machinery of OBM-Net as well as its ability to generalize at inference time to differences in (a) the number of actual objects, (b) the number of hypothesis slots and (c) the number of observations. Next, we evaluate the performance of OBM-Net on an image domain in which the underlying observation space is substantially different from the hypothesis space. Finally, we evaluate the performance of OBM-Net on the complex simulated household robot domain shown in Figure 1, and validate the ability of OBM-Net to capture an object with underlying dynamics and complex properties, as well as its utility for downstream robotics object-fetching tasks. We provide additional evaluation of our approach in Sections A, B and C.

**Baselines and Metrics**    In each domain, we compare OBM-Net to online learned baselines of LSTM (Hochreiter and Schmidhuber, 1997) and set transformer (Lee et al., 2018) (details in E.3), as well as to task-specific baselines. All learned network architectures are structured to use $\sim 50000$

| Model | Online | Learned | Normal | Elongated | Mixed | Angular | Noise |
|---|---|---|---|---|---|---|---|
| OBM-Net | + | + | **0.157** | **0.191** | **0.184** | **0.794** | **0.343** |
| Set Transformer | + | + | 0.407 | 0.395 | 0.384 | 0.794 | 0.424 |
| LSTM | + | + | 0.256 | 0.272 | 0.274 | 0.799 | 0.408 |
| VQ | + | - | 0.173 | 0.195 | 0.191 | 0.992 | 0.947 |
| Set Transformer | - | + | 0.226 | 0.248 | 0.274 | **0.816** | **0.406** |
| Slot Attention | - | - | 0.254 | 0.267 | 0.268 | 0.823 | 0.504 |
| K-means++ | - | - | **0.103** | **0.139** | **0.135** | 0.822 | 1.259 |
| GMM | - | - | 0.113 | 0.141 | 0.136 | 0.865 | 1.207 |

Table 1: **Quantitative Results on Online Clustering.** Comparison of performance on clustering performance across different distributions. Reported error is the L2 distance between predicted and ground truth means. Methods in the bottom half of table operate on observations in bulk and thus are not directly comparable.

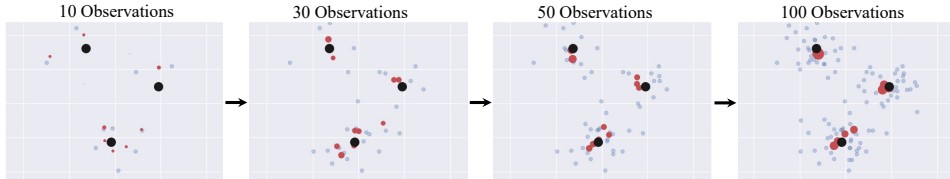

Figure 3: **Qualitative Visualization of OBM-Net.** Illustration of OBM-Net execution on the *Normal* distribution setting. Decoded value of hypothesis (with size corresponding to confidence) shown in red, with ground truth clusters in black. Observations are shown in blue.

parameters. Unless otherwise noted, models except OBM-Net are given and asked to predict the ground truth number of components $K$, while OBM-Net uses 10 hypothesis slots. Results are reported in terms of MSE error $\frac{1}{K} \min_j \|y_k - m_j\|$ (with respect to the most confident $K$ hypotheses for OBM-Net).

## 5.1 ONLINE CLUSTERING

**Setup.** To check the basic operation of the model and understand the types of problems for which it performs well, we first test our approach on simple clustering problems with the same input and output spaces, but different types of data distributions, each a mixture of three components. We train on 1000 problems with observation sequences of length 30 drawn from each problem distribution and test on 5000 problems from the same distribution. In every case, the means of the three components are drawn at random for each problem. We consider a set of five problem distributions, a *Normal* setting in which each component is a 2D Gaussian with identical variance across individual dimensions and components, *Enlongated* and *Mixed* settings where 2D Gaussians have more variation across different components and *Angular* and *Noise* settings where underlying distributions are non-Gaussian in nature. We provide precise details about distributions in Section E.1.

**Baselines and Metrics.** In addition to the online learned baselines described in Section 5, we compare our approach with following task specific clustering methods: *Batch, non-learning*: K-means++ (Arthur and Vassilvitskii, 2007) and expectation maximization (EM) (Dempster et al., 1977) on a Gaussian mixture model (SciKit Learn implementation); *Online, non-learning*: vector quantization (Gray, 1984); *Batch, learning*: set transformer (Lee et al., 2018) and slot attention (Locatello et al., 2020).

**Results.** We compare our approach to each of the baselines for the five problem distributions in Table 1. The results in this table show that on *Normal*, *Mixed*, and *Elongated* tasks, OBM-Net performs better than learned and constructed online clustering algorithms, but does slightly worse than offline clustering algorithms. Such discrepancy in performance is to be expected due to the fact that OBM-Net is running and being evaluated online. On the *Angular* and *Noise* tasks, OBM-Net is able to learn a useful metric for clustering and outperforms both offline and online alternatives.

Next, we provide a qualitative illustration of execution of OBM-Net on the *Normal* clustering task in Figure 3 as a trajectory of observations are seen. We plot the decoded values of hypothesis slots in red, with size scaled according to confidence, and ground-truth cluster locations in black. OBM-Net is able to selectively refine slot clusters to be close to ground truth cluster locations even with much longer observation sequences than it was trained on. We provide qualitative visualization of individual modules of OBM-Net in Section A.2 as well as performance on increased numbers of clusters in Section A.4. We further provide ablations of each proposed component of OBM-Net in Section A.3.

**Generalization.** We next assess the ability of OBM-Net to generalize at inference time to differences in the number of input observations as well as differences in the underlying number of hypothesis slots used on the *Normal* distribution. On the left side of Figure 4, we plot the error of

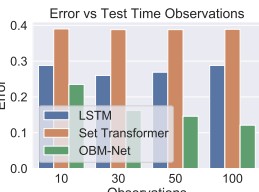

| Model | Slots | Ground Truth Clusters | | |
|---|---|---|---|---|
| | | 3 | 5 | 7 |
| OBM-Net | 10 | **0.162** | 0.214 | 0.242 |
| | 20 | 0.175 | **0.195** | 0.213 |
| | 30 | 0.188 | 0.197 | **0.205** |
| Set Transformer | - | 0.261 | 0.279 | 0.282 |
| Vector Quantization | - | 0.171 | 0.199 | **0.205** |

Figure 4: **(left) Generalization with Increased Observations.** Plot of LSTM, Set Transformer and OBM-Net errors when executed *at test time* on different number of observations from the *Normal* distribution. With increased observations, OBM-Net error continues to decrease while other approaches obtain higher error. **(right) Generalization with Different Hypothesis Slots.** Error of OBM-Net, when executed *at test time* with a different number of hypothesis slots on test distributions with different numbers of ground true components. In all cases, OBM-Net is trained on 3-component problems with 10 slots. OBM-Net achieves good performance with novel number of hypothesis slots, and outperforms different instances of the Set Transformer trained with the ground truth number of cluster components as well as vector quantization.

LSTM, Set Transformer, and OBM-Net as a function of the number observations seen at inference time. We find that when OBM-Net is given more observations then seen during training time (all models are trained with observations of length 30), it is able to further improve its performance, while both LSTM and Set Transformer results suffer. We believe that such generalization ability is due to the inductive bias added to OBM-Net, and provide an analysis in Section A.3. We provide additional analysis of this generalization across all distributions in Table A6 and find similar results.

On the right side of Figure 4, we investigate the generalization ability of OBM-Net at *inference time* to increases in both the number of hypothesis slots and the underlying number of mixture components from which observations are drawn. We compare to the set transformer and to VQ, both of which *know the correct number of components at inference time*. We find that OBM-Net generalizes well to increases in hypothesis slots, and exhibits improved performance with large number of underlying components, performing comparably to or better than the VQ algorithm. We further note that none of the *learning* baselines can adapt to different numbers cluster components at inference time, but find that OBM-Net outperforms the set transformer even when it is trained on the ground truth number of clusters in the test. We provide additional generalization analysis in Section A.1.

## 5.2 IMAGE-BASED DOMAINS

We next validate the ability of OBM-Net to perform entity monitoring on image inputs, which requires OBM-Net to synthesize a latent representation for slots, and learn to perform association, update, and transition operations in that space.

**Setup.** We experiment with two separate image-based domain, each consisting of a set of similar entities (2D digits or 3D airplanes). We construct entity monitoring *problems* by selecting $K$ objects in each domain, with the desired $y$ values being images of those objects in a canonical viewpoint. An input observation sequence is generated by randomly selecting one of those $K$ objects, and generating an observation $z$ corresponding to a random viewpoint of the object. Our two domains are: (1) **MNIST**: Each object is a random image in MNIST, with observations corresponding to rotated images, and the desired outputs being the un-rotated images; (2) **Airplane**: Each object is a random object from the Airplane class in ShapeNet (Chang et al., 2015), with observations corresponding to airplane renderings (using Blender) at different viewpoints and the desired outputs the objects rendered in a canonical viewpoint. We provide details in Section E.1 and use $K = 3$ components.

**Baselines.** In addition to our learned baselines, we compare with a task specific baseline, batch K-means, in a latent space that is learned by training an autoencoder on the observations. In this setting, we were unable to train the Set Transformer stably and do not report results for it.

**Results.** In Table 2, we find that our approach significantly outperforms other comparable baselines in both accuracy and generalization. We further visualize qualitative predictions from our model in Figure 5. We find that our highest confidence decoded slots correspond to ground truth objects.

## 5.3 SIMULATED HOUSEHOLD ROBOT DOMAINS

Finally, we validate that OBM-Net can solve the entity monitoring task in simulated robotic settings.

**Setup.** We model a robot moving within a house, as pictured in Figure 1, in the PyBullet simulation environment. In this house, each problem will involve following a trajectory consisting of a sequence of 50 locations. These locations are distributed across 5-6 separate rooms, with later locations potentially revisiting earlier locations. At each location, the robot looks around and if there is a table

| Model | Learned | MNIST | | | | Airplanes | | | |
|---|---|---|---|---|---|---|---|---|---|
| Observations | | 10 | 30 | 50 | 100 | 10 | 30 | 50 | 100 |
| OBM-Net | + | **7.143** | **5.593** | **5.504** | **5.580** | **4.558** | **4.337** | **4.331** | **4.325** |
| LSTM | + | 9.980 | 9.208 | 9.166 | 9.267 | 5.106 | 4.992 | 4.983 | 4.998 |
| K-means | + | 13.596 | 12.505 | 12.261 | 12.021 | 7.246 | 6.943 | 6.878 | 6.815 |

Table 2: **Quantitative Results on Image Domain.** Comparison of entity-monitoring performance on MNIST and Airplane datasets across 10, 30, 50, 100 observations. For OBM-Net, LSTM and K-means we use a convolutional encoder/decoder trained on the data. We train models with 30 observations and report MSE error.

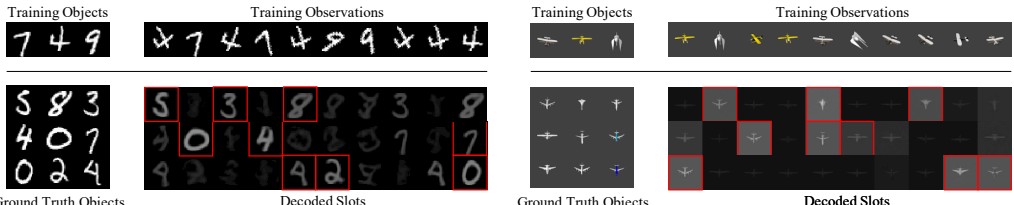

Figure 5: **Qualitative Visualization of OBM-Net Execution on Images.** Qualitative visualization of two image-based association tasks (left: MNIST, right: airplanes). At the top of each is an example training problem, illustrated by the true objects and an observation sequence. Each of the next rows shows an example test problem, with the ground truth objects and decoded slot values. The three highest-confidence hypotheses for each problem are highlighted in red, and correspond to ground-truth objects.

within view (which happens about 50% of the time), it will get an observation of one of the objects on the table or an empty observation otherwise. Each new problem has 8 tables whose locations are drawn from a larger set of potential table locations and on each table there will be two objects drawn from a small set of classes, e.g. lamp, cushion, etc. Each object class has a characteristic stochastic movement pattern, with one object class sequentially teleporting between tables (details in appendices). The goal is for the robot to be able to construct hypotheses for each distinct object it has seen and to be able to predict for each object the table it is currently on and its location relative to the table. More precisely, the input sequence of observations $z$ corresponds to a segmented depth map of a single object visible given the camera pose at a particular location in the trajectory (or an empty observation in the case no object is visible), as well as which table it is resting on and its positional offset relative to the table. The desired output $y$ values are, for each object seen so far, is the predicted table $y^t$ it is on currently as well its associated offsets relative to the predicted table, $y^o$.

We train on a total of 10000 randomly sampled trajectories in the same floor plan, but with new randomly drawn object instances and tables for each trajectory. We test using 1000 trajectories, with test object meshes drawn from a set *disjoint from* the set of object meshes used during training (but sharing the same semantic class). To test the flexibility of the approach, we consider three different configurations of object classes on tables, with the motion pattern of each of the 3 object classes illustrated in Section E.1, as well additional setup details and example observations.

**Metrics.** To test the efficacy of our approach, we measure to what extent each hypothesis slot $m_i$ can recover both the table that the associated object is on, as well as the object's position relative to the table. We match a hypothesis slot $k$ with each object label $y_i$ by computing $\arg\min_k \|y_i^o - m_k^o\| + \text{Loss}_{\text{CE}}(y_i^t - m_k^t)$. For each match, we report the accuracy of $m_k^t$ matching $y_i^t$, and as well the mean absolute error between $y_i^o$ and $m_k^o$. When the table prediction for $y_i$ is incorrect, we set mean absolute error to be equal to half the table size (0.15), as reported table offsets are meaningless in that case. In this setting, both OBM-Net and associated baselines use 10 hypothesis slots.

**Baselines.** In addition to our learned baselines, we compare to two task specific baselines. We construct a simple clustering baseline for this problem. Given a localized input-segmented depth map, we extract object offsets by averaging all points in the point cloud associated with each segment. To associate objects dynamically across time, we use batch K-means clustering on the inferred object candidate offsets and associated table identities to obtain a set of objects. We further compare OBM-Net with the more complex spatial-temporal clustering method used in the STRANDS project (Hawes et al., 2017) to infer objects in a real robotic setup from our underlying segmented depth maps, as well as a hand-crafted DAF system using ground truth dynamics. For all learned models, we convert the segmented depth maps into downsampled 3D pointclouds.

**Results.** Table 3 shows that OBM-Net outperforms the baselines in both estimating the supporting tables and regressing the relative position of the objects across different number of observations. Figure 6 (left) shows the prediction error of all methods as a function of the number of steps since the robot last saw an object; observe that OBM-Net is substantially better at long-term memory than the LSTM and set transformer, and still outperforms the clustering and STRANDS baselines even

| Model | Learned | Configuration A | | | | | | Configuration B | | | | | | Configuration C | | | | | |
|---|---|---|---|---|---|---|---|---|---|---|---|---|---|---|---|---|---|---|---|
| | | Table Accuracy | | | Position Error | | | Table Accuracy | | | Position Error | | | Table Accuracy | | | Position Error | | |
| Observations | | 10 | 25 | 50 | 10 | 25 | 50 | 10 | 25 | 50 | 10 | 25 | 50 | 10 | 25 | 50 | 10 | 25 | 50 |
| OBM-Net | + | **0.984** | **0.926** | **0.809** | **0.019** | **0.041** | **0.078** | **0.989** | **0.924** | **0.795** | **0.021** | **0.046** | **0.082** | **0.988** | **0.932** | **0.873** | **0.027** | **0.052** | **0.080** |
| Set Transformer | + | 0.883 | 0.619 | 0.476 | 0.034 | 0.066 | 0.089 | 0.919 | 0.771 | 0.542 | 0.024 | 0.052 | 0.093 | 0.885 | 0.745 | 0.649 | 0.037 | 0.056 | 0.089 |
| LSTM | + | 0.839 | 0.661 | 0.406 | 0.058 | 0.093 | 0.126 | 0.875 | 0.716 | 0.514 | 0.053 | 0.094 | 0.123 | 0.892 | 0.717 | 0.519 | 0.052 | 0.091 | 0.130 |
| Clustering | - | 0.761 | 0.695 | 0.485 | 0.053 | 0.070 | 0.103 | 0.761 | 0.695 | 0.488 | 0.053 | 0.070 | 0.103 | 0.761 | 0.695 | 0.488 | 0.053 | 0.069 | 0.103 |
| STRANDS | - | 0.900 | 0.733 | 0.610 | 0.033 | 0.057 | 0.085 | 0.940 | 0.841 | 0.737 | 0.023 | 0.048 | 0.087 | 0.973 | 0.832 | 0.774 | 0.031 | 0.055 | 0.086 |
| DAF | - | 0.959 | 0.807 | 0.670 | 0.022 | 0.043 | 0.081 | 0.937 | 0.871 | 0.787 | 0.021 | 0.039 | 0.084 | 0.974 | 0.914 | 0.803 | 0.030 | 0.053 | 0.083 |

Table 3: **Quantitative Analysis of OBM-Net on Simulated Household Domain.** Quantitative comparison of OBM-Net with baselines across 3 studied household domain configurations across 10, 25, 50 observations.

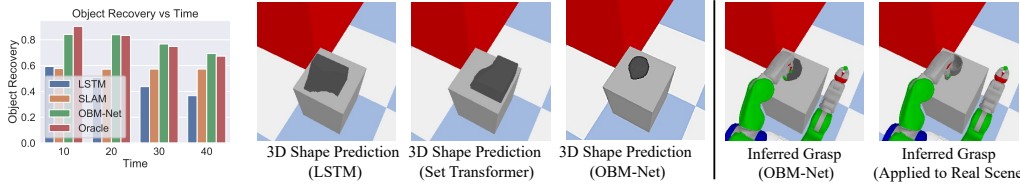

3D Shape Prediction (LSTM)    3D Shape Prediction (Set Transformer)    3D Shape Prediction (OBM-Net)    Inferred Grasp (OBM-Net)    Inferred Grasp (Applied to Real Scene)

Figure 6: **(left) Object Recovery over Time.** Percentage of objects correctly recovered as a function of timesteps since seeing the object last. OBM-Net performs similarly to an oracle with ground truth dynamics. **(middle) 3D Reconstructions.** Illustration of 3D reconstructions of hypothesis from each model. OBM-Net obtains accurate 3D reconstructions. **(right) Estimated Grasps.** We utilize the predicted 3D mesh from OBM-Net to infer a grasp which successfully enables the grasp of a real object in the ground truth scene.

with long inter-observation gaps. As an upper bound, we compare with an oracle model, which knows ground truth object identity and dynamics (ignoring object collision). We find that OBM-Net performs similarly to the oracle model (performance across all models drops due to stochasticity), and in some cases does better, perhaps by taking account object collisions.

By adding a shape occupancy prediction head (Mescheder et al., 2019) to OBM-Net, we can also regress the underlying 3D shapes of our objects. We predict each shape at $32 \times 32 \times 32$ resolution, decoding each occupancy at each voxel coordinate using a MLP head conditioned on a hypothesis state. Quantitatively, we find that our approach gets 95.33% accuracy compared to 72.74% accuracy obtained by a LSTM and 73.67% obtained by a set transformer when predicting voxels for each test mesh in the test set. We provide visualization of predicted shapes from OBM-Net in Figure A4.

**Object fetching.** Finally, we verify that object hypotheses from OBM-Net can usefully support a task in which a robot has to retrieve an object it has previously observed. First, we consider the task of finding a previouly-encountered object. We train LSTM, set transformer, and OBM-Net to predict underlying object class $y^c$ for each object hypothesis, as well as shape estimate and location. Given a desired object class (for example, either a plant, cushion, or bucket in configuration A)) we wish to find, the robot examines each prediction $(y_i, c_i)$ and navigates in the simulated world to look for an object of the specified class, based on predictions of $y_i^t$ and $y_i^o$. We measure the number of predictions that need to be queried to find the object, as well as a overall success percentage of trials in which the robot succeeded within 10 attempts. On this task, we find that a LSTM obtains an overall planning success of 68.75% with an average number of 5.38 hypotheses investigated before finding an object. In contrast, the set transformer obtains a planning success of 81.25% with on average 4.88 attempts. We find that OBM-Net performs best and is able to find the object of the desired class 100% of the time, with an average of 2.03 hypotheses examined before finding the object.

Next, we qualitatively analyze the 3D reconstructions of each object hypothesis and its ability to support manipulation. Given a 3D reconstruction, we compute grasps on the underlying shape by looking for parallel planar surfaces large enough to accommodate the gripper. We then try to execute that grasp on the target 3D object we wish to grasp in the (simulated) real world. As illustrated in Figure 6, we find that the 3D reconstruction of object hypotheses from OBM-Net is accurate enough to enable grasping of a real 3D shape. In contrast, predictions from LSTM and set transformer baselines are significantly poorer and do not enable downstream manipulation.

**Discussion** This work has demonstrated that using algorithmic bias inspired by a classical solution to the problem of filtering to estimate the state of multiple objects simultaneously, coupled with modern machine-learning techniques, we can arrive at solutions that learn to perform and generalize well. Importantly, the same underlying system, with *no prior knowledge* about the types of observations or desired output hypotheses or the frequency of observations, is able to learn to perform data-association and state estimation to solve a variety of entity monitoring problems as well as to support an object-based memory system for a robot in a dynamically changing environment.

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

# Appendix

To fully understand the underlying machinery of OBM-Net, we provide additional qualitative and quantitative verification of OBM-Net on online clustering in Section A, on dynamic clusters in Section B and in simulated household domains in Section C. We provide a derivation of the sparsity loss utilized in training in Section D. Finally, we provide experimental and architecture details of OBM-Net in Section E.

## A ONLINE CLUSTERING RESULTS

We provide quantitative and qualitative results on online clustering to further analyze and understand the underlying performance and algorithmic computation performed by OBM-Net. First, we analyze the generalization of OBM-Net to increased number of clusters of inputs in Section A.1. Next, we provide qualitative visualization of OBM-Net in Section A.2, and ablations in Section A.3. We further analyze performance on problems with a larger number of clusters in Section A.4.

### A.1 GENERALIZATION

We provide additional analysis of the ability of OBM-Net to *generalize at test time* to increased number of observations and an increased number of clusters.

**Generalization to Increased Observation Number.** We report performance of different models across different distributions in Table A6 when executed *at test time* with an increased number of observations. We find that OBM-Net is able to obtain better performance with increased number of observations across all different distributions, while other learned baselines perform worse. Furthermore, we find that OBM-Net outperforms other learned baselines in all distributions except for rotation. For rotation we find that when training with 10,000 different distributions, OBM-Net exhibits better performance of 0.555 compared to Set Transformer performance of 0.647 and LSTM performance of 0.727 with 30 observations (and similarly outperforms Set Transformer and LSTM at larger number of observations).

**Inferring Object Number.** In contrast to other algorithms, OBM-Net learns to predict both a set of object properties $y_k$ of objects and a set of confidences $c_k$ for each object. This corresponds to the task of both predicting the number of objects in a set of observations, as well as the associated object properties. We evaluate the ability of OBM-Net to regress object number at *test time* in scenarios where the number of objects (underlying clusters) is different than that of training. We evaluate on the *Normal* distribution with a variable number of component distributions, and measure inferred components through a threshold confidence. OBM-Net is trained on a dataset with 3 underlying components. We find in Figure A1 that OBM-Net is able to infer the presence of more component distributions (as they vary from 3 to 6), with improved performance when cluster centers are sharply separated (right figure of Figure A1).

### A.2 QUALITATIVE VISUALIZATION

**Submodule Visualization.** We find that individual modules learned by OBM-Net are interpretable. We visualize the attention weights of each hypothesis slot in Figure A2 and find that each hypothesis slot learns to attend to a local region next to the value it decodes to. We further visualize a plot of relevance weights in Figure A3 across an increasing number of observations where individual observations are drawn distributions with different levels of noise with respect to cluster centers. We find that as we see more observations, the relevance weight of new observations decreases over time, indicating that OBM-Net learns to pay the most attention towards the first set of observations it sees. In addition, we find that in distributions with higher variance, the relevance weight decreases more slowly, as later observations are now more informative in determining cluster centers.

### A.3 ABALATIONS

We ablate each component of OBM-Net and present results in Table A1 on the *Normal* distribution. We test removing $\mathcal{L}_{\text{sparse}}$ (sparsity), removing learned slot embeddings (learned memory) — where

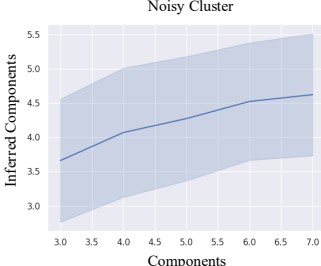
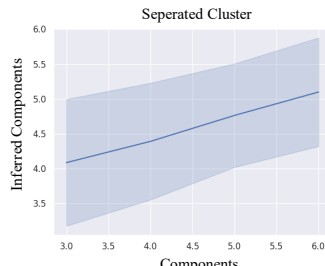

Figure A1: **Generalization to Increased Cluster Number.** Plots of inferred number of components using a confidence threshold in OBM-Net compared to the ground truth number of clusters (OBM-Net is trained on only 3 clusters). We consider two scenarios, a noisy scenario where cluster centers are randomly drawn from -1 to 1 (left) and a scenario where all added cluster components are well seperated from each other (right). OBM-Net is able to infer more clusters in both scenarios, with better performance when cluster centers are more distinct from each other.

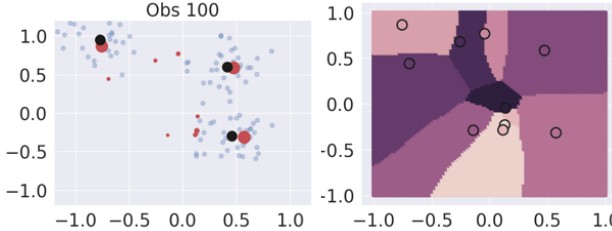

Figure A2: **Visualization of Attention Weights.** Plot of decoded values of slots (in red) with confidence shown by the size of dot (left), and what slot each input assigns the highest attention towards (right) (each slot is colored differently, with assigned inputs colored in the same way). Note alignment of regions on the right with the decoded value of a slot on the left.

Figure A3: **Visualization of Relevance Weights.** Plots of the magnitude of relevance weights with increased observation number on different distributions with differing standard deviation (noise).

instead, in individual hypothesis slots, we store the explicit values of inputs, removing the **suppress** modules (suppression) and removing the **relevance** module (relevance). We find that each of our proposed components enables better performance on the underlying clustering task. Interestingly, we further find that the addition of **relevance** enables our approach to generalize at test time to larger numbers of observations.

| Sparsity | Learned Memory | Supression | Relevance | Observations | | | |
|---|---|---|---|---|---|---|---|
| | | | | 10 | 30 | 50 | 100 |
| − | − | − | − | 0.382 | 0.452 | 0.474 | 0.487 |
| + | − | − | − | 0.384 | 0.412 | 0.423 | 0.430 |
| + | + | − | − | 0.335 | 0.357 | 0.366 | 0.387 |
| + | + | + | − | 0.279 | 0.274 | 0.278 | 0.282 |
| + | + | + | + | **0.238** | **0.157** | **0.137** | **0.131** |

Table A1: **Abalation Analysis.** We ablate each component of OBM-Net on the *Normal* distribution . When learned memory is ablated, OBM-Net updates states based on observed values (appropriate in the *Normal* distribution dataset).

## A.4 LARGER NUMBER OF CLUSTERS

We measure the performance of OBM-Net when trained with a large number of clusters and slots. We utilize the *Normal* distribution setting, but generate underlying training input observations from a total of 30 difference components, and train OBM-Net with a total of 30 slots. We train OBM-Net with 50 observations, and measure performance at inferring cluster centers with between 50 to 100 observations. We report performance in Table A2 and find that OBM-Net obtains good performance

| Model | Online | Observations | | | |
|---|---|---|---|---|---|
| | | 50 | 65 | 80 | 100 |
| OBM-Net | + | **0.158** | **0.154** | **0.151** | **0.147** |
| VQ | + | 0.162 | 0.157 | 0.153 | 0.148 |
| K-means++ | - | **0.155** | **0.151** | **0.148** | **0.146** |
| GMM | - | 0.156 | 0.151 | 0.149 | 0.147 |

Table A2: **Performance on Large Number of Clusters.** Comparison of performance on *Normal* distribution, when underlying distributions have a large number of components. We use 30 components, and train models with 50 observations. Each cluster observation and center is drawn between -1 and 1, with reported error as the L2 distance between predicted and ground truth means.

in this setting, out-performing the strong online baseline VQ, and performing similarly to K-means++ which directly operates on all input observations at once.

## B  DYNAMIC DOMAINS

We further verify the ability of OBM-Net to perform entity monitoring in a dynamic setting and compare its performance with that of a classical data-association filter.

**Setup.** We evaluate performance of dynamic entity monitoring using moving 2D objects. A *problem* involves a trajectory of observations $z$ of the $K$ dynamically moving objects, with desired $y$ values being the underlying object positions. Objects evolve under a linear Gaussian dynamics model, with a noisy observation of a single object observed at each step (details in Section E.1). This task is typical of tracking problems considered by DAF. All learning-based models are trained on observation sequences of length 30. To perform well in this task, a model must discover that it needs to estimate the latent velocity of each object, as well as learn the underlying dynamics and observation models. We utilize $K = 3$ for our experiments.

**Baselines.** We compare with the de-facto standard method, Joint Probabilistic Data Association (JPDA) (Bar-Shalom et al., 2009), which uses hand-built ground-truth models (serving as an oracle). We further compare with our learned online baselines of Set Transformer (Lee et al., 2018) and LSTM (Hochreiter and Schmidhuber, 1997) methods.

**Result.** Quantitative performance, measured in terms of prediction error on true object locations, is reported in Table A3. We can see that the Set Transformer cannot learn a reasonable model at all. The LSTM performs reasonably well for short (length 30) sequences but quickly degrades relative to OBM-Net and JPDA as sequence length increases. We note that OBM-Net performs comparably to, but just slightly worse than, JPDA. *This is strong performance because OBM-Net is generic and can be adapted to new domains given training data without the need to hand-design the models in JPDA.* We believe that these gains are due to the inductive biases built into our architecture.

| Model | Observations | | | |
|---|---|---|---|---|
| | 10 | 20 | 30 | 40 |
| OBM-Net | **0.415** | 0.395 | 0.382 | 0.394 |
| Set Transformer | 0.699 | 0.701 | 0.854 | 1.007 |
| LSTM | 0.422 | 0.400 | 0.445 | 0.464 |
| JPDA (oracle) | 0.683 | **0.372** | **0.362** | **0.322** |

Table A3: **Performance on Dynamic Objects.** Comparison of different methods on estimating the state of 3 dynamically moving objects. All learning models are trained with 1000 sequences of 30 observations. We report MSE error. JPDA uses the ground-truth observation and dynamics models.

## C  HOUSEHOLD DOMAINS RESULTS

We provide additional analysis of the ability of OBM-Net to perform entity monitoring in a simulated household domain.

**Additional Simulated Household Robot Results.** We present qualitative visualizations of 3D shapes obtained by OBM-Net when executed on test trajectories. In Figure A4 we illustrate different

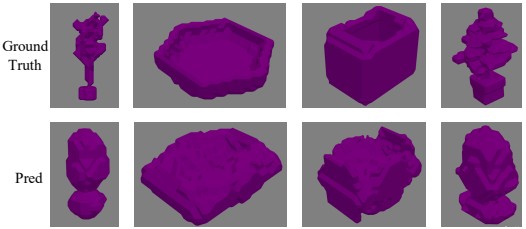

Figure A4: **3D Reconstructions of OBM-Net.** Illustration of predicted 3D shapes from OBM-Net when OBM-Net is executed on a test trajectory. We further visualize ground truth meshes seen in the trajectory and find that predicted shapes coarsely match ground truth meshes.

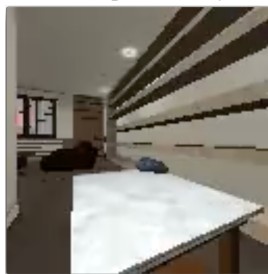
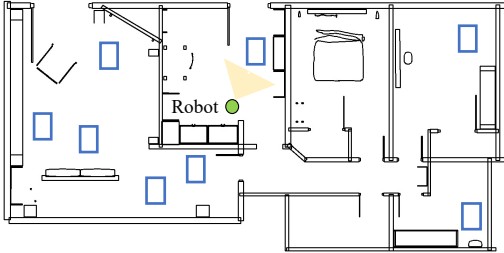

Figure A5: **Illustration of iGibson Environment. (left)** Illustration of example RGB input in our iGibson environment. **(right)** Example configuration of tables in our iGibson environment (tables drawn in blue).

predicted 3D shapes with their associated ground truth 3D shape. We find that reconstructions from OBM-Net appear to coarsely match the underlying shape of ground truth 3D meshes.

**iGibson Environment.** We further validate the efficacy of OBM-Net on the more complex household domain of iGibson. We illustrate an example input observation of our environment, and the corresponding configuration of tables and objects in Figure A5. We utilize the same configuration settings as the Pybullet environment, training models on trajectories of length 50, consisting of 8 tables with 2 objects on them each. We utilize object classes and movement patterns from Configuration A described in Section 5.3 in the main paper. We provide additional dataset details in Section E.1. We compare OBM-Net with LSTM and clustering baselines discussed in Section 5.3 of the main paper. We use the same metrics as described in Section 5.3. In Table A4 we report results on this household setting. We find that OBM-Net performs significantly better than LSTM and Clustering baselines.

**Scanned iGibson Environment.** We further validate the efficacy of OBM-Net on utilizing real house scans from Figure A5 as the background of our environment. We utilize the same configuration settings as the Pybullet environment, training models on trajectories of length 50, consisting of 8 tables with 2 objects on them each. We utilize object classes and movement patterns from Configuration A described in Section 5.3 in the main paper. We provide additional dataset details in Section E.1. We compare OBM-Net with LSTM and clustering baselines discussed in Section 5.3 of the main paper. We use the same metrics as described in Section 5.3. In Table A5 we report results on this household setting. We find that OBM-Net performs significantly better than LSTM and Clustering baselines.

| Model | Learned | Table Accuracy | | | Position Error | | |
|---|---|---|---|---|---|---|---|
| Observations | | 10 | 25 | 50 | 10 | 25 | 50 |
| OBM-Net | + | **0.992** | **0.925** | **0.813** | **0.159** | **0.234** | **0.301** |
| LSTM | + | 0.883 | 0.625 | 0.489 | 0.203 | 0.294 | 0.354 |
| Clustering | - | 0.798 | 0.638 | 0.554 | 0.204 | 0.266 | 0.318 |

Table A4: **Quantitative Results on iGibson.** Comparison of performance of OBM-Net and baselines on the iGibson environment.

| Model | Learned | Table Accuracy | | | Position Error | | |
|---|---|---|---|---|---|---|---|
| Observations | | 10 | 25 | 50 | 10 | 25 | 50 |
| OBM-Net | + | **0.970** | **0.889** | **0.782** | **0.153** | **0.232** | **0.296** |
| LSTM | + | 0.891 | 0.628 | 0.499 | 0.195 | 0.297 | 0.344 |
| Clustering | - | 0.802 | 0.632 | 0.576 | 0.212 | 0.258 | 0.304 |

Table A5: **Quantitative Results on Scanned Gibson Houses.** Comparison of performance of OBM-Net and baselines on scanned Gibson environment.

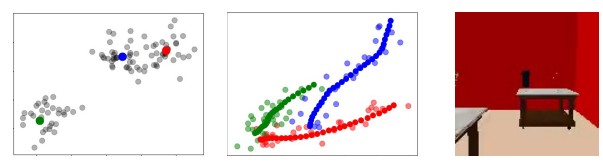 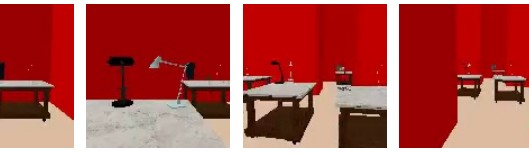

Gaussian Domain    Dynamic Domain           Robotic Domain

Figure A6: **Qualitative Visualization of Domains.** Visualizations of the Normal Gaussian, Dynamic domains and Robotic domains. Observations are transparent while ground truth states are bolded for gaussian and dynamic domains. Four sample image observations shown for robotic domain.

## D    Sparsity Loss

In this section, we show that $\mathcal{L}_{\text{sparse}}(\mathbf{c})$ encourage confidences $\mathbf{c}$ to be sparse. Recall that

$$\mathcal{L}_{\text{sparse}}(\mathbf{c}) = -\log\|\mathbf{c}\| \quad . \tag{2}$$

where $\|\mathbf{c}\|$ is the L2 norm which is a convex function. Recall that $c$, the confidence vector, defines a polyhedron, since it is the set of points that are non-negative, and whose elements sum up to one. The maximum of a convex function over a polyhedra must occur at the vertices, which correspond to an assignment of 1 to a single $c_i$ and 0s to every other value of $\mathbf{c}$. Next we consider the minimum of $\|c\|$ given that its elements sum up to one. This is equivalent to finding the stationary points of the Lagrangian $L(c, \lambda)$

$$L(c, \lambda) = \sum_i c_i^2 + \lambda(\sum_i c_i - 1) \tag{3}$$

By taking the gradient of the above expression, we find that the stationary value corresponds to each $c_i$ being equal. Since the function is convex, this corresponds to the minimum of $\|c\|$. Thus $\mathcal{L}_{\text{sparse}}(\mathbf{c})$ is maximized when each individual confidence is equal.

## E    Experimental Details

In this section, we provide details of our experimental approach. We first discuss the details of datasets used in Section E.1. Next, we provide the model architectures used in Section E.2. Finally, we provide details on the baselines we compare with in Section E.3.

### E.1    Dataset Details

We first provide detailed experimental settings for each of the datasets considered in the paper.

**Online Clustering.**    In online clustering, we utilize observations drawn from the following distributions, where Gaussian centers are drawn uniformly from -1 to 1.

1. *Normal*: Each 2D Gaussian has standard deviation 0.2. The normal setting is illustrated in Figure A6.
2. *Mixed*: Each distribution is a 2D Gaussian, with fixed identical variance across each individual dimension, but with the standard deviation of each distribution drawn from a uniform distribution from (0.04, 0.4).
3. *Elongated*: Each distribution is a 2D Gaussian, where the standard deviation along each dimension is drawn from a uniform distribution from (0.04, 0.4), but fixed across distributions.
4. *Angular*: Each distribution is a 2D Gaussian with identical standard deviation across dimension and distribution, but points above $\pi$ are wrapped around to $-\pi$ and points below $-\pi$ wrapped to $\pi$. Gaussian means are selected between $(-\pi, -2\pi/3)$ and between $(2\pi/3, \pi)$. The standard deviation of distributions is $0.3 * \pi$.
5. *Noise*: Each distribution has 2 dimensions parameterized by Gaussian distributions with standard deviation 0.5, but with the values of the remaining 30 dimensions drawn from a uniform distribution from $(-1, 1)$.

**Dynamic Domains.**    Next, in the dynamics domain, we implement our dataset using the StoneSoup library[*]. We initialize the location of each cluster from a Gaussian distribution with standard deviation

---

[*]https://stonesoup.readthedocs.io/en/v0.1b3/stonesoup.html

Figure A7: **Object Dynamics in Household Domain.** Illustration of object dynamics in our simulated household environment. Objects exhibit either vertical, horizontal, or teleportation motion, dependent on the object class across configurations A, B, C. We illustrate the motion of each object class, with start and end images corresponding to 20 timesteps of motion of a object, except for objects that teleport – these change at each timestep.

1.5 and initialize velocity in each directory from a Gaussian distribution with standard deviation of 0.02. At each timestep, Gaussian noise is added to velocities with magnitude 1e-4. We show example tracks in Figure A6. Our JPDA implementation is also from the StoneSoup library.

**Image Domains.** In the image domain, for MNIST, we use the 50000 images in the training set to construct the training problems, and the 10000 images in the non-overlapping test set to construct the test problems. For the Airplane dataset, we use 1895 airplanes to construct the training problems, and 211 different airplanes to construct the test problems. Each airplane is rendered with 300 viewpoints.

**Robotics Domains.** For the robotics domains, we implement our embodied house environment in Pybullet, and construct a house with $x$ and $y$ axis between $-1$ and $1$. We utilize furniture assets from (Xia et al., 2020) for each of the individual classes of objects considered, with 50% of the objects (sorted alphabetically) being used for the training dataset and the 50% of the objects used for the test dataset. Each individual object is scaled by a factor of 0.1 to fit on each table. Each table has size 0.15 by 0.1 in our setting. Our constructed house environment uses the floor plan illustrated in Figure 1 of the main paper. Across configurations, objects, which each individual step of the trajectory corresponding to a $1/60$ of second advancement of simulation time in PyBullet. In each configuration, objects in separate object classes move with velocities of $(0.6, 0.0)$, $(0.0, 0.6)$ and $(0.3, 0.3)$ units per second respectively. In the presence of collision, all objects involved in the collision event stop moving, making the underlying dynamics of objects a stochastic process. We illustrate example images observations from our environment in Figure A6 (though segmented depth maps of objects are instead input to our model) and illustrations of the underlying dynamics of objects in Figure A7. Objects which teleport, teleport at each timestep (provided there is no collision).

For the iGibson house, we utilize the Pomaria environment in iGibson environment. This house has $x$ and $y$ axis roughly between $-4$ and $4$, and thus we scale tables to a size of 0.45 by 0.3 in the environment, and proportionally scale up the size of individual objects as well as their underlying movement speed. To sample trajectories in both settings, we sample a set of points across rooms in a house and utilize motion planning to infer paths connecting each individual point.

## E.2 MODEL/BASELINE ARCHITECTURES

We provide the overall architecture details for the LSTM in Figure A8a, for the Set Transformer in Figure A8b and OBM-Net in Figure A9a. For image experiments, we provide the architecture of the encoder in Figure A10a and decoder in Figure A10b. Both LSTM, OBM-Net, and autoencoding baselines use the same image encoder and decoder. For robotics experiments, we provide the architecture of the shape decoder in Figure A9b.

In OBM-Net memory, the function update$(s_k, n_k, e)$ is implemented by applying a 2 layer MLP with hidden units $h$ which concatenates the vectors $s_k, n_k, e$ as input and outputs a new state $u_k$ of dimension $h$. The value $n_k$ is encoded using the function $\frac{1}{1+n_k}$, to normalize the range of input to be between 0 and 1. The function attend$(s_k, n_k, e)$ is implemented in an analogous way to update, using a 2 layer MLP that outputs a single real value for each hypothesis slot.

For the function relevance$(s_k, n_k, e)$, we apply $\text{NN}_1$ per hypothesis slot, which is implemented as a 2 layer MLP with hidden units $h$ that outputs a intermediate state of dimension $h$. $(s_k, n_k, e)$ is fed into $\text{NN}_1$ in an analogous manner to update. $\text{NN}_2$ is applied to average of the intermediate representations

of each hypothesis slot and is also implemented as a 2 layer MLP with hidden unit size $h$, followed by a sigmoid activation. We use the ReLU activation for all MLPs. $NN_3$ is represented is GRU, which operates on the previous slot value.

## E.3 BASELINE DETAILS

All baseline models are trained using prediction slots equal to the ground truth of components. To train the Set Transformer to act in an online manner, we follow the approach in (Santoro et al., 2018) and we apply the Set Transformer sequentially on the concatenation of an input observation with the current set of hypothesis slots. Hypothesis slots are updated based off the new values of the slots after applying self-attention (Set Transformer Encoder). Hypothesis slots are updated based off the new values of the slots after applying self-attention (Set Transformer Encoder). We use the Chamfer loss to train baseline models, with confidence set to 1.

| Dense → h |
| --- |
| Dense → h |
| LSTM(h) |
| Dense → h |
| Dense → output |

(a) The model architecture of the LSTM baseline. The hidden dimension $h$ used is 96 for synthetic Gaussian distributions and 128 for Image datasets. For image experiments, the first 2 and last 2 fully connected layers are replaced with image encoders and decoders.

| Dense → h |
| --- |
| Dense → h |
| Set Transformer Encoder |
| Set Transformer Decoder |

(b) The model architecture of the Set Transformer baseline. The hidden dimension $h$ is 48 for the synthetic Gaussian distributions. We use the encoder and decoder defined in (Lee et al., 2018) with 4 heads and hidden dimension $h$.

Figure A8: Architecture of baseline models.

| Dense → h |
| --- |
| Dense → h |
| OBM-Net Memory |
| Dense → h |
| Dense → output |

(a) The model architecture of OBM-Net. The hidden dimension $h$ is 64 is for synthetic Gaussian distributions and 128 for the image/robotics experiments. For image experiments, the first and last 2 linear layers are replaced with convolutional encoders and decoders.

| $(x, y, z)$ → Dense → h |
| --- |
| Concat$(h, \text{state})$ |
| Dense → h |
| Dense → 1 |

(b) The shape decoder of OBM-Net used in the robotics experiments. The shape decoder takes as input a voxel coordinate as well as a slot value and predicts a occupancy for the voxel.

Figure A9: Architecture of OBM-Net and the shape decoder.

| 5x5 Conv2d, 32, stride 2, padding 2 |
| --- |
| 3x3 Conv2d, 64, stride 2, padding 1 |
| 3x3 Conv2d, 64, stride 2, padding 1 |
| 3x3 Conv2d, 64, stride 2, padding 1 |
| 3x3 Conv2d, 128, stride 2, padding 1 |
| Flatten |
| Dense → h |

(a) The model architecture of the convolutional encoder for image experiments.

| Dense → 4096 |
| --- |
| Reshape $(256, 4, 4)$ |
| 4x4 Conv2dTranspose, 128, stride 2, padding 1 |
| 4x4 Conv2dTranspose, 64, stride 2, padding 1 |
| 4x4 Conv2dTranspose, 64, stride 2, padding 1 |
| 4x4 Conv2dTranspose, 64, stride 2, padding 1 |
| 3x3 Conv2d, 3, stride 1, padding 1 |

(b) The model architecture of the convolutional decoder for image experiments.

Figure A10: The model architecture of the convolutional encoder and decoder for image experiments.

| Type | Model | Online | Observations | | | |
|---|---|---|---|---|---|---|
| | | | 10 | 30 | 50 | 100 |
| Normal | OBM-Net | + | 0.235 | 0.162 | 0.146 | 0.128 |
| | Set Transformer | + | 0.390 | 0.388 | 0.388 | 0.389 |
| | LSTM | + | 0.288 | 0.260 | 0.269 | 0.288 |
| | VQ | + | 0.246 | 0.172 | 0.147 | 0.122 |
| | Set Transformer | + | 0.295 | 0.261 | 0.253 | 0.247 |
| | K-means++ | - | 0.183 | 0.107 | 0.086 | 0.066 |
| | GMM | - | 0.189 | 0.118 | 0.087 | 0.067 |
| Mixed | OBM-Net | + | 0.255 | 0.184 | 0.164 | 0.147 |
| | LSTM | + | 0.306 | 0.274 | 0.284 | 0.290 |
| | Set Transformer | + | 0.415 | 0.405 | 0.407 | 0.408 |
| | VQ | + | 0.262 | 0.192 | 0.169 | 0.145 |
| | Set Transformer | - | 0.309 | 0.274 | 0.266 | 0.261 |
| | K-means++ | - | 0.206 | 0.135 | 0.105 | 0.088 |
| | GMM | - | 0.212 | 0.136 | 0.105 | 0.079 |
| Enlongated | OBM-Net | + | 0.258 | 0.192 | 0.173 | 0.161 |
| | LSTM | + | 0.314 | 0.274 | 0.288 | 0.300 |
| | Set Transformer | + | 0.394 | 0.391 | 0.394 | 0.394 |
| | VQ | + | 0.265 | 0.194 | 0.172 | 0.149 |
| | Set Transformer | - | 0.309 | 0.244 | 0.240 | 0.232 |
| | K-means++ | - | 0.213 | 0.139 | 0.113 | 0.092 |
| | GMM | - | 0.214 | 0.141 | 0.112 | 0.086 |
| Rotation | OBM-Net | + | 0.892 | 0.794 | 0.749 | 0.736 |
| | LSTM | + | 0.799 | 0.796 | 0.795 | 0.794 |
| | Set Transformer | + | 0.793 | 0.794 | 0.782 | 0.782 |
| | VQ | + | 0.956 | 1.000 | 1.000 | 0.984 |
| | Set Transformer | - | 0.815 | 0.784 | 0.779 | 0.772 |
| | K-means++ | - | 0.827 | 0.834 | 0.823 | 0.802 |
| | GMM | - | 0.842 | 0.875 | 0.867 | 0.848 |
| Noise | OBM-Net | + | 0.375 | 0.343 | 0.338 | 0.334 |
| | LSTM | + | 0.419 | 0.406 | 0.405 | 0.407 |
| | Set Transformer | + | 0.434 | 0.424 | 0.425 | 0.424 |
| | VQ | + | 1.479 | 0.948 | 0.826 | 0.720 |
| | Set Transformer | - | 0.436 | 0.407 | 0.398 | 0.394 |
| | K-means++ | - | 1.836 | 1.271 | 1.091 | 0.913 |
| | GMM | - | 1.731 | 1.215 | 1.056 | 0.856 |

Table A6: **Generalization with Increased Observations.** Error of different models when executed *at test time* on different number of observations across different distributions. We train models with 3 components and 30 observations.

