# OpenReview forum: "Learning an Object-Based Memory System"
_ICLR.cc/2022/Conference — ICLR 2022 Submitted_

### Official Review · Reviewer_qPet · 2021-11-02

**Correctness:** 4
**Technical Novelty And Significance:** 2
**Empirical Novelty And Significance:** 1
**Recommendation:** 3
**Confidence:** 4

**Main Review:**

In general, the paper is easy to follow, the evaluation proves some potential of the proposed method. However, there are some critical concerns of the paper listed below:

A.	This paper addressed the problem “entity monitoring”. Based on the problem formulation, it is very related to the object re-identification and instance retrieval problem. How does it differ? I would expect some highlights on this regard in the introduction and related works.

B.	All experiments are performed with synthetic dataset. This does not expose much real-world challenges. Section 5.2 only evaluates on a single object category for 3D objects, i.e.  the airplane. However, often the problem at hand requires the method to handle multiple instances of multiple categories. In an indoor environment, you can easily find multiple chairs in a room together with many other categories of objects. Ref 1) below performed a similar evaluation, but with the full 40 categories of ModelNet40.  Moreover, another critical point from my view is the lack of evaluation on real-world dataset. One can extract the image crops corresponding to each object captured from different viewpoints from some public 3D scene dataset, e.g. ScanNet (http://www.scan-net.org/), as how it is done in Ref 2) below.

C.	In line with the above points, there are methods that are closely related but are not covered in the discussion and in the comparison. Some recent related works are listed below for references:
1)	Bai, Song, et al. "Re-ranking via metric fusion for object retrieval and person re-identification." Proceedings of the IEEE/CVF Conference on Computer Vision and Pattern Recognition. 2019.
2)	Bansal, Vaibhav, Gian Luca Foresti, and Niki Martinel. "Where Did I See It? Object Instance Re-Identification With Attention." Proceedings of the IEEE/CVF International Conference on Computer Vision. 2021.
3)	He, Shuting, et al. "Transreid: Transformer-based object re-identification." ICCV 2021.
4)	Zhang, Yifu, et al. "Fairmot: On the fairness of detection and re-identification in multiple object tracking." International Journal of Computer Vision (2021): 1-19.
5)	Zhang, Xiao, et al. "Refining Pseudo Labels with Clustering Consensus over Generations for Unsupervised Object Re-identification." Proceedings of the IEEE/CVF Conference on Computer Vision and Pattern Recognition. 2021.
6)	Huang, Tsung-Wei, et al. "Multi-View Vehicle Re-Identification using Temporal Attention Model and Metadata Re-ranking." CVPR Workshops. Vol. 2. 2019.

**Summary Of The Paper:**

This paper addresses the “entity monitoring” problem. They propose a neural-network architecture that uses self-attention as a mechanism for data association to construct a memory of the objects in the environment and demonstrate its effectiveness in a set of illustrative problems.

**Summary Of The Review:**

To summaries, the current submission does not situate the method well in the literature because some closely related works are not covered in the paper. The evaluation is also not convincing in terms of how the method can handle real-world challenges and lacks evaluation of real-world data. I therefore vote for a rejection.

---

> ### Author Response · Authors · 2021-11-13
> **[Pre-Revision] Reviewer qPet Response**
>
> Thank you for your constructive criticism. Below, we provide clarifications of each of the listed concerns, as well as state our planned changes (and experiments that we will run). Please let us know if our planned revisions will address your concerns.
>
> **Q1) Differences with Prior Work**
>
>  Please see our main response.
>
> **Q2) Realistic Images**
>
> We agree that it would be desirable to run our experiments on data from the real world. However, we note that the problem we wish to solve is to estimate the underlying state of objects across time -- not to re-identify objects across different views.
>
> In terms of near realistic results, we actually run our approach in simulated photorealistic household environments from iGibson in Appendix Section C. In this rebuttal, we will further run an additional comparison in our simulated robotic setup utilizing scanned real iGibson houses (corresponding to real images similar to ScanNet).
>
> **Q3) Related Work**
>
> We will add each of these works to the related work of our paper. However, as noted in the main author response, we attempt to solve a substantially different problem from object reidentification.

---

> > ### Comment · Reviewer_qPet · 2021-11-23
> > **Not fully convinced by the argument**
> >
> > Thanks authors for the response, and adding more related work discussion would be good.
> >
> > However, I could not agree with the argument that the "entity monitoring" is different from object re-identification. Both problems aim to obtain a representation or underlying state of the object in order to make the association when the object is observed across different time and/or views. Both problem could be the case when the object is moving, or the observer is moving, or both the object and observer are moving.
> >
> > Simulations using observations rendered with CAD models (perfect geometric capturing) or 3D scans (imperfect geometric capturing, e.g holes in the reconstruction), cannot not fully represent images captured from real-world scenes.
> > To demonstrate the usefulness of the method, experiment with real-world images is necessary, and the dataset is available for such evaluation, e.g. ScanNet.

---

> > > ### Author Response · Authors · 2021-11-23
> > > **Clarification of Differences**
> > >
> > > We would like to clarify that there are really substantial differences between the problem settings.
> > >
> > > Re-identification
> > > - input is a large batch of images, with bounding boxes for object detections
> > > - output is an assignment of object detections to unique IDs
> > > - generally specific to an object category
> > > - loss is based on the correctness of assignments to IDs
> > >
> > > Entity monitoring
> > > - input is a stream of object detections
> > > - system operates *online* at every step producing a set of hypothesized objects
> > > - object state (not just position) can change over time
> > > - agnostic to input and output type/encoding, which can in general be in very different spaces
> > > - loss is based on accuracy of output representations and number of hypotheses
> > >
> > > It is true that much of the work on object re-id focuses on designing algorithms for learning representations that map the same objects into nearby points so that associations can be determined reliably. In our work, that is implicitly part of the job of the latent encoding---but in the end we don't care about the associations and instead focus on producing an output representation that specified by the training data. So, for example, we could train an OBM-net to produce estimates of an object's mass or material, or how full a container is. And, given the architectural design, as in an EM-type clustering method, there is no hard assignment of detections to hypotheses and we would not be able to recover identification information for the re-id problem.
> > >
> > > The primary contributions of this paper are
> > > - the problem formulation
> > > - the design of an architecture and loss functions that allow the system to learn to be a generic data-association filter, learning the necessary models for novel versions of the problem
> > > - a demonstration that the architecture and losses are effective in a variety of different types of problems
> > >
> > > It is true that it would have been nice to have demonstrated this in a setting with real RGBD image sequences collected in a non-stationary domain, used to produce rich aggregated output representations of the objects' states. However, there does not exist a real dataset for accomplishing this task, and gathering such a dataset is a huge undertaking that is well beyond the scope of this initial paper.

---

> > > > ### Comment · Reviewer_qPet · 2021-11-26
> > > > **Final discussion**
> > > >
> > > > While the description of Object Re-ID provided by the authors match most of the settings whose main goal is to learn a more distinctive object representation, there also exist methods that integrate such module for long-term tracking with videos as the input. The latter could either directly perform learn and update the model in an online manner, like the seminar work TLD [1], or learn the representation offline with videos as the input, like the recent FairMOT [2].
> > > >
> > > > Based on the description of the problem setting of the "entity monitoring", it is perhaps more related to long-term object tracking. I would suggest authors to be more aware of the related problems, so the proposal can be better positioned in the literature.
> > > >
> > > > At the moment, I don't see clearly the novelty, and I don't see how it respond to real-world challenges, thus I am still for rejection.
> > > >
> > > > [1] Tracking-Learning-Detection, Zdenek Kalal, et al, TPAMI2010
> > > > [2] FairMOT: On the Fairness of Detection and Re-Identification in Multiple Object Tracking,
> > > > Yifu Zhang, Chunyu Wang, Xinggang Wang, Wenjun Zeng, Wenyu Liu, IJCV2021

---

> ### Author Response · Authors · 2021-11-20
> **[Post-Revision] Reviewer qPet Response**
>
> Thank you for your constructive feedback. We have made substantial modifications to our paper following your review.
>
> - We have discussed the individual differences of our approach and that of object reidentification in the introduction of the paper.
> - We have added an experiment applying our approach to scanned houses from iGibson in appendix C.
> - We have added a discussion with approaches for object reidentification in the related work of our paper, and discussed how our approach is different from such approaches.
>
> As the discussion period is about to end, please don’t hesitate to let us know if there are any additional clarifications that we can offer, as we would love to convince you of the merits of the paper. Thanks!

---

### Official Review · Reviewer_V7qP · 2021-11-02

**Correctness:** 3
**Technical Novelty And Significance:** 2
**Empirical Novelty And Significance:** 2
**Recommendation:** 3
**Confidence:** 4

**Main Review:**

The idea of encoding "algorithmic priors" into deep learning architectures has been explored in the past for various problems and usually is a sound idea and provides benefits to aspects such as complexity, efficacity, or performance of the resulting method over a pure end-to-end learning-based method. As such, the idea proposed in the paper is sensible and has merit.

Despite this, there are several issues with the paper in its current state. One of them is the apparent disconnect between the data association based description and motivation in the beginning, and the clustering focused experimental validation. While it might be reasonable to consider certain aspects to be clustering-like, these similarities should be described and motivated. There also appears to be no comparison to any DAF method to be presented.

The description of the method itself, while understandable at a high level, lacks detail and specifics. The method seems to utilize various deep learning "building blocks" such as slots, attention, etc., without introducing what they are or how their function captures algorithmic aspects. The capturing of algorithmic aspects is another problematic part of the method description, as the algorithm that is being reproduced is never actually described or shown. Therefore, the reader is left wondering what aspects are being replicated and why.

The experiments are very disconnected from the paper's motivation, and their link to the initially described task is unclear. The second experiment on MNist and planes also seems to be more about the presence or absence of rotational invariance of the employed embeddings, more so than the method itself. Making the comparison confusing and purpose unclear. Another problem with the clustering based evaluation is the metric being used. L2 or similar distance metrics are simply not useful metrics when evaluating clustering results and thus a widely accepted metric such as normalized mutual information would be appropriate to evaluate these results. The last experiment, which seems to be more in the direction of the initially described problem does not appear to have an actual DAF method included in the baselines. Given that the proposed method seemed to be replicating a specific DAF method I would have expected to see at least that method as a baseline.

**Summary Of The Paper:**

The paper proposes an end-to-end system for the data association and filtering (DAF) problem. The architecture is built to mimic components typically found in DAF systems to provide a sort of algorithmic prior to the network. The resulting system is evaluated on several different tasks using synthetic data.

**Summary Of The Review:**

The general idea of replicating traditional algorithm designs in a deep learning architecture is sensible the lack of detail in the method description and unclear experimental design and evaluation means that the paper is not in a publishable state at this stage.

---

> ### Author Response · Authors · 2021-11-13
> **[Pre-Revision] Reviewer V7qP Response**
>
> Thank you for your constructive criticism. Below, we provide clarifications of each of the listed concerns, as well as state our planned changes (and experiments that we will run). Please let us know if our planned revisions will address your concerns.
>
> **Q1) DAF Comparisons**
>
> We actually provided a comparison to a DAF in Appendix Section B of the paper on dynamic domains in the initial submission of the paper. We further plan to add a comparison to DAF in the household domain -- but directly comparing with DAF can be problematic and difficult due to the fact that both observation and dynamic functions must be explicitly coded and are quite difficult to specify in this domain.
>
> **Q2) Descriptions of Method**
>
> We will provide additional clarification on how each of our proposed architectural blocks correspond to the underlying algorithmic computation done in DAF. In particular, our use of separate slots corresponds to a set of hypotheses in a DAF system. The use of attention corresponds to computation of the compatibility of each hypothesis with a given new observation p(s_new | obs, s_old).
>
> **Q3) Experiments Don’t Match Problem Setting**
>
> Please see general response.

---

> > ### Comment · Reviewer_V7qP · 2021-11-26
> > **Reply**
> >
> > Thank you for the clarification and improvements to the paper. While the changes improve the paper and address some of the issues, the overall clarity of the problem setup and link between idea, method, and experiments is still insufficient. It is apparent that all reviewers have slightly different interpretations of what this paper describes and neither matches what the authors want to convey. This I believe is the core issue which despite the additions to the paper is not resolved.
> >
> > As stated in my review, while I can see how clustering can capture some of the aspects and properties of the proposed method it also poses problems with being a very specific problem with its own expectations. Given that the paper draws inspiration from DAF literature it might be possible to find a good experimental setup in that literature. To some extent, the proposed method is also related to general state estimation, though with more varied temporal aspects than the typical state estimation performed in robotics.
> >
> > Finally, while I might increase my score slightly, the overall recommendation would still sit on the side of a rejection as, despite the improvements to the paper, the core issues have not been resolved.

---

> ### Author Response · Authors · 2021-11-20
> **[Post-Revision] Reviewer V7qP**
>
> Thank you for your constructive feedback. We have made substantial modifications to our paper following your review.
>
> - We have added a comparison with DAF in Table 3. We find that our approach slightly outperforms are hand-crafted DAF despite the latter knowing the ground truth transition dynamics of objects
> - We have clarified how each individual computation block relates to the computation done in a DAF algorithm.
> - We have clarified in the introduction the underlying purpose of each individual experiment in the paper.
>
> As the discussion period is about to end, please don’t hesitate to let us know if there are any additional clarifications that we can offer, as we would love to convince you of the merits of the paper. Thanks!

---

### Official Review · Reviewer_nrGQ · 2021-11-03

**Correctness:** 3
**Technical Novelty And Significance:** 2
**Empirical Novelty And Significance:** 2
**Recommendation:** 3
**Confidence:** 4

**Main Review:**

The entity-monitoring problem as formulated in the paper does not seem to pose the challenges outlined in the intro. Post reading the intro I was expecting to see something akin to really long-term tracking with associations across objects even when there are huge gaps in temporal coherence in environments with significant distractors. In contrast, the problem setup focuses on very limited settings with fairly low complexity.

"In particular, we train a system to construct a memory of the objects in the environment, without prior models of the robot’s sensing, the types of objects to be encountered, or the patterns in which they might move in the environment. "

I am not sure what without prior models of the robot's sensing means. Since the model is being trained on the agent's input. It is just a question of when the learning is happening (online vs. offline). Since the training is being done online, one of the things that is happening is learning a feature representation for clustering/slotting similar objects together. Why does this feature representation need to be learnt online? Why can we not use a pre-trained representation trained in a self supervised manner with observations collected from the robot. Once we have such a representation how would proposed method compare to just clustering observations on a pre-trained representation. For instance in the qualitative examples shown in Figure 5 I would be surprised if unsupervised pre-trained representations cannot make the same association.

From an experimental stand point all the tasks seem fairly non-standard due to definition of the task. There are other datasets like CATER https://rohitgirdhar.github.io/CATER/ which setup versions of the problem. Why setup a different problem rather than use CATER or some of the long term tracking benchmarks https://github.com/wangdongdut/Long-term-Visual-Tracking which involve object re-association?

From an architecture and modeling standpoint there does not seem to be significantly different from https://arxiv.org/pdf/2006.15055.pdf. Moreover, the paper does not evaluate on the same benchmark tasks Locatello et.al evaluate on. What is the reason for this? Evaluating on the same benchmarks would allow comparison of differences in modeling details more effectively since the evaluation OBM-Net is empirical.

**Summary Of The Paper:**

The paper defines an entity-monitoring problem where the goal is to identify the distinct objects see in an episode where the agent/model moves through the scene/observes partial state. The paper proposes the OBM-Net model architecture to address this problem and identify all the distinct objects observed over each episode. The key idea of OBM-Net to have fixed set of slots which allow tracking multiple hypothesis over time and use an attention mechanism to update the slots with observations over time.

**Summary Of The Review:**

The entity-monitoring problem as evaluated in the paper does not pose the challenges outlined in the intro. I am not convinced that the proposed method is a better alternative for associating/clustering objects compared to using pre-trained representations on data from the corresponding domains.

---

> ### Author Response · Authors · 2021-11-13
> **[Pre-Revision] Reviewer nrGQ Response**
>
> Thank you for your constructive criticism. Below, we provide clarifications of each of the listed concerns, as well as state our planned changes (and experiments that we will run). Please let us know if our planned revisions will address your concerns.
>
> **Q1) Tasks Don’t Match the Entity Monitoring Problem**
>
> Please see our main response.
>
> **Q2) No Prior Models of Robot Sensing**
>
> In many robotic systems, approaches utilize carefully engineered transition and error models of input observations to accurately estimate an underlying probabilistic state of the surrounding world. These models are generally very difficult to acquire and tune.  In this paper, we do not utilize such hand engineered systems, and thus do not rely on prior models of robot sensing.
>
> We indeed explored utilizing a pretrained representation of object state (obtained from autoencoding) compared to one utilized by OBM-Net. We found, however, that such an approach leads to very poor performance (12.21 MSE error with 100 observations on MNIST compared 5.580 of OBM-Net). We posit that it is crucial for our latent state to be a representation amenable to downstream data association, as opposed a fixed pretrained representation.  In particular, in Figure 5, notice that we are not trying to make **associations** of images to form a cluster center -- rather we are attempting to consolidate the information across different images of the same object together to predict the underlying state of an object (corresponding to rendering the object at a new pose). We clarify this in the experiments section.
>
> **Q3) Nonstandard Experimental Setting**
>
> While it is true that the problem we wish to solve is different from prior work, the entity monitoring problem we propose is a crucial one that must be solved for a robot agents to operate in long-horizon open-world domains. In such domains, a robot will encounter a variety of objects over the course of execution, and must aggregate and remember the underlying states of individual objects so that it may find and manipulate these objects for subsequent downstream tasks. Please see our general response for additional discussion.
>
>
> **Q4) Differences with Slot Attention**
>
> Our underlying training objective and modeling outputs are substantially different than that of slot attention. The slot attention architecture aims to predict a set of objects given a single static image while OBM-Net seeks to output a set of hypothesis and associated confidences given a sequence of temporal observations. We will clarify these differences in the related work. We attempted to modify the slot attention architecture for our setting and found that it performed significantly worse than OBM-Net. We will further add these numbers in our paper revision.

---

> ### Author Response · Authors · 2021-11-20
> **[Post-Revision] Reviewer nrGQ Response**
>
> Thank you for your constructive feedback. We have made substantial modifications to our paper following your review.
>
> - We have clarified in the introduction the underlying purpose of individual tasks we consider
> - We have clarified what we mean by no prior model of robot sensing -- we mean we do not know the error distribution of input observations.
> - We have clarified in the introduction and related work the differences between our approach and prior work
> - We have added a comparison to slot attention in Table 1 in the paper. As stated in the pre-revision comment, our approach significantly outperforms slot attention on clustering, as their approach is constructed to focus on unsupervised object segmentation instead of temporal filtering to estimate states of the world.
>
> As the discussion period is about to end, please don’t hesitate to let us know if there are any additional clarifications that we can offer, as we would love to convince you of the merits of the paper. Thanks!

---

### Author Response · Authors · 2021-11-13
**[Pre-revision] General Response to all Reviewers**

We thank reviewers for their constructive comments. However, we would like to clarify the underlying task we are trying to solve, which we believe reviewers may have misunderstood. We have also left individual comments to each reviewer about proposed revisions to address weaknesses they brought.

# 1. Scope

We are focused on the problem of entity monitoring. In the entity monitoring problem, we are interested in aggregating information across observations across time to obtain a low variance estimate of the underlying **states** of various objects in the world as well as their associated **confidences**. Solving this problem is of critical importance to developing robots that can operate in long-horizon open-world domains.  The goal of this paper is both to illustrate the problem and its importance as well as to provide baselines and a first solution.  As far as we know, there are no pre-existing testing domains or algorithms that address this important problem domain.

Crucially, this problem is substantially different from both “long-term tracking” (Reviewer nrGQ) and “object reidentification” (Reviewer qPet). The problems studied in both these tasks focus on rediscovering an instance of a particular object which has been seen at previous times, but *not* on the problem of aggregating and estimating the underlying state of individual objects across a period of temporally sparse observations, which is the focus of our paper (reviewer qPet). Existing datasets (Reviewer nrGQ) do not focus on estimating such long term states of the world.

To enable close study of our algorithm’s ability to estimate the individual states of different objects, we chose to first consider the problem of cluster center estimation (Reviewer V7qP). The problem of cluster center estimation represents the simplest instantiation of an entity monitoring task, as we must learn to aggregate past observations together to obtain a coherent estimate of the underlying state (locations) of individual clusters. While, in principle, there exist other metrics for evaluating clustering, we choose to measure the accuracy of regressed cluster centers, as this state estimation problem corresponds to that of the entity monitoring problem we wish to study.

Next we considered our method’s ability to estimate  the underlying state of objects when observations are drawn in a more high dimensional visual domain (Reviewer V7qP). We construct both MNIST and airplanes problems to test OBM-Net’s ability to aggregate information from a visual domain together to estimate the underlying state of different objects. In particular, in the airplane domain, for OBM-Net to successfully reconstruct an airplane in a canonical pose, it must learn to aggregate information from all prior views together (as each individual view does not contain sufficient information to reconstruct the full airplane) and to do so without a continuous, temporally dense, observation sequence that would enable tracking..

Finally, we study the ability of OBM-Net to estimate the state of objects in complex simulated household domains where objects move over time as described in the introduction of the paper. We study this in Section 5.3 of the paper on simulated Pybullet environments as well in photorealistic simulated houses in Appendix Section C. This setting is especially substantially different than either “long-term tracking” and “object reidentification” (Reviewer nrGQ, qPet) as we focus on estimating the **state** of objects, corresponding to their 3D shape, position and associated table identity.

We will revise both our introduction and related works sections to emphasize these points more strongly.

---

### Author Response · Authors · 2021-11-20
**[Post-revision] General Response to all Reviewers**

We thank reviewers for their thoughtful comments.

The focus of our work is to construct a system which can estimate and remember the underlying **state** of objects in the world, such as their underlying shape, identity, position, and numerosity, to enable a mobile robot to remember and interact with objects in a dynamically changing environment. Solving this task is crucial to enable effective mobile manipulators, and is substantially different from the closest prior work of object reidentification, which only focuses on re-identifying instances of particular objects across time. We provide additional discussion of this point in the pre-revision response to reviewers.

We believe we have addressed each of the concerns brought up by each individual reviewer. We have revised the manuscript with the following changes:

- We have clarified the differences between our work and object reidentification in the introduction
- We have clarified the purpose of individual studied experiments in the introduction
- We have added related work on object reidentification and further discussed differences
- We have discussed how the construction of our network is inspired by the form of a DAF
- We have a comparison to slot attention in Table 1
- We have a comparison to a hand-constructed DAF system with ground truth knowledge of data transitions in Table 3.
- We have evaluated our approach on real images from the scanned iGibson environment in appendix C

To enable better visibility of changes, we have highlighted each individual change in blue.

---

### Decision · Program_Chairs · 2022-01-20

**Decision:**

Reject

**Comment:**

This paper proposed a long-term object-based memory system for robots.  The proposed method builds on existing ideas of data association filters and neural-net attention mechanisms to learn transition and observation models of objects from labelled trajectories.  The proposed method was compared with baseline algorithms in a set of experiments.

The initial reviews raised multiple concerns about the paper. Reviewers nrGQ and  V7qP commented on the conceptual gap between the problem proposed in the introduction and the extent of the experiments. Reviewer qPet understood the paper to be a form of object re-identification and was concerned about the limited comparisons with related work.  The author response clarified their goal of estimating the states of the objects in the world, which they state is different from the goals of long-term tracking and object reidentification mentioned by the reviewers.  The authors also clarified the relationship to other work in slot-attention and data association filters.

The ensuing discussion among the reviewers indicated that the paper's contribution remained unclear even after the author response. Two reviewers noted the paper did not clearly communicate the problem being solved (all reviewers had a different view of the problem in the paper).  These reviewers wanted a better motivation for the problem being addressed in this paper.  The third reviewer remained unconvinced that the problem in the paper was different from long-term object tracking.

Three knowledgeable reviewers indicate reject as the contributions of the paper were unclear to all of them. The paper is therefore rejected.